# Cezanne (OTUD7B) regulates HIF-1α homeostasis in a proteasome-independent manner

Anja Bremm[1,2,*], Sonia Moniz[3], Julia Mader[1], Sonia Rocha[3,**] & David Komander[2,***]

## Abstract

The transcription factor HIF-1α is essential for cells to rapidly adapt to low oxygen levels (hypoxia). HIF-1α is frequently deregulated in cancer and correlates with poor patient prognosis. Here, we demonstrate that the deubiquitinase Cezanne regulates HIF-1α homeostasis. Loss of Cezanne decreases HIF-1α target gene expression due to a reduction in HIF-1α protein levels. Surprisingly, although the Cezanne-regulated degradation of HIF-1α depends on the tumour suppressor pVHL, hydroxylase and proteasome activity are dispensable. Our data suggest that Cezanne is essential for HIF-1α protein stability and that loss of Cezanne stimulates HIF-1α degradation via proteasome-independent routes, possibly through chaperone-mediated autophagy.

**Keywords** Cezanne; HIF-1α; hypoxia; Lys11-linked ubiquitin chains; ubiquitin

**Subject Categories** Post-translational Modifications, Proteolysis & Proteomics; Signal Transduction

## Introduction

Cellular adaptation to hypoxia depends on the heterodimeric transcription factor HIF. The α-subunit of HIF is carefully regulated in order to prevent inappropriate target gene expression, and dysfunction of this pathway is associated with various diseases [1,2]. Rapid turnover of HIF-1α in normoxia is mediated by a well-characterised oxygen-dependent enzymatic cascade involving prolyl hydroxylases (PHDs) and a specialised cullin–RING E3 ubiquitin (Ub) ligase complex (CRL2[VHL] [3]) that consists of the tumour suppressor von Hippel–Lindau (pVHL), cullin-2, elongin B/C and the small RING finger protein RBX1. Hydroxylation of HIF-1α by PHDs results in its recognition by pVHL, ubiquitination, and rapid degradation by the proteasome. Deregulated HIF-1α levels caused by inactivating *VHL* mutations predispose humans to a variety of cancers, in which the regulation

machinery for HIF-1α degradation has been studied extensively [4]. In hypoxia, oxygen-dependent hydroxylases are gradually inhibited, and HIF-1α protein levels rise substantially [5,6]. More recently, oxygen-independent degradation mechanisms of HIF-1α have been described [7–9], emphasising the complexity of HIF-1α homeostasis.

Deubiquitinases (DUBs) oppose the function of E3 ligases by hydrolysing Ub chains [10]. Ub-specific protease (USP) 20 binds pVHL [11] and was shown to deubiquitinate and stabilise HIF-1α [12]. Recent reports also suggested roles for other USP DUBs in HIF-1α regulation [13–15].

Here, we demonstrate that the Lys11 linkage-specific ovarian tumour (OTU) DUB Cezanne regulates HIF transcriptional activity by directly affecting HIF-1α protein homeostasis in a proteasome-independent way.

## Results and Discussion

### Cezanne regulates HIF-1α-dependent gene expression

OTU DUBs control many important cell signalling pathways [16]. To test whether OTU family members are involved in regulating HIF-1α transcriptional activity in hypoxia, 14 human OTU DUBs were depleted in U2OS cells, and reporter gene assays were performed. USP20 served as a positive control [12] and its depletion reduced HIF-1α activity as suggested by the literature (Supplementary Fig S1A). Interestingly, knockdown of Cezanne-1/OTUD7B (hereafter referred to as Cezanne) but not of any other OTU DUB (including OTUD7A/Cezanne-2) decreased HIF-1α activity to the same extent as knockdown of USP20 (Supplementary Fig S1A).

To validate the results from the siRNA screen, Cezanne was depleted by a siRNA pool or by three individual siRNA oligonucleotides, which decreased HIF-1α transcriptional activity after exposure to hypoxia (Fig 1A, Supplementary Fig S1D). Knockdown of Cezanne was efficient at the mRNA level (Fig 1B, Supplementary Fig S1B) and at the protein level as examined with an antibody raised against the Cezanne OTU domain (Fig 1C, Supplementary Fig S1C). Loss of Cezanne increased transcriptional activity of NF-κB (Supplementary Fig S1E) as shown

1   Buchmann Institute for Molecular Life Sciences, Institute of Biochemistry II, Goethe University, Frankfurt (Main), Germany
2   Medical Research Council Laboratory of Molecular Biology, Cambridge, UK
3   College of Life Sciences, Centre for Gene Regulation and Expression, University of Dundee, Dundee, UK
    *Corresponding author. Tel: +49 69 79842510; E-mail: bremm@em.uni-frankfurt.de
    **Corresponding author. Tel: +44 1382 385792; E-mail: s.rocha@dundee.ac.uk
    ***Corresponding author. Tel: +44 1223 267160; E-mail: dk@mrc-lmb.cam.ac.uk

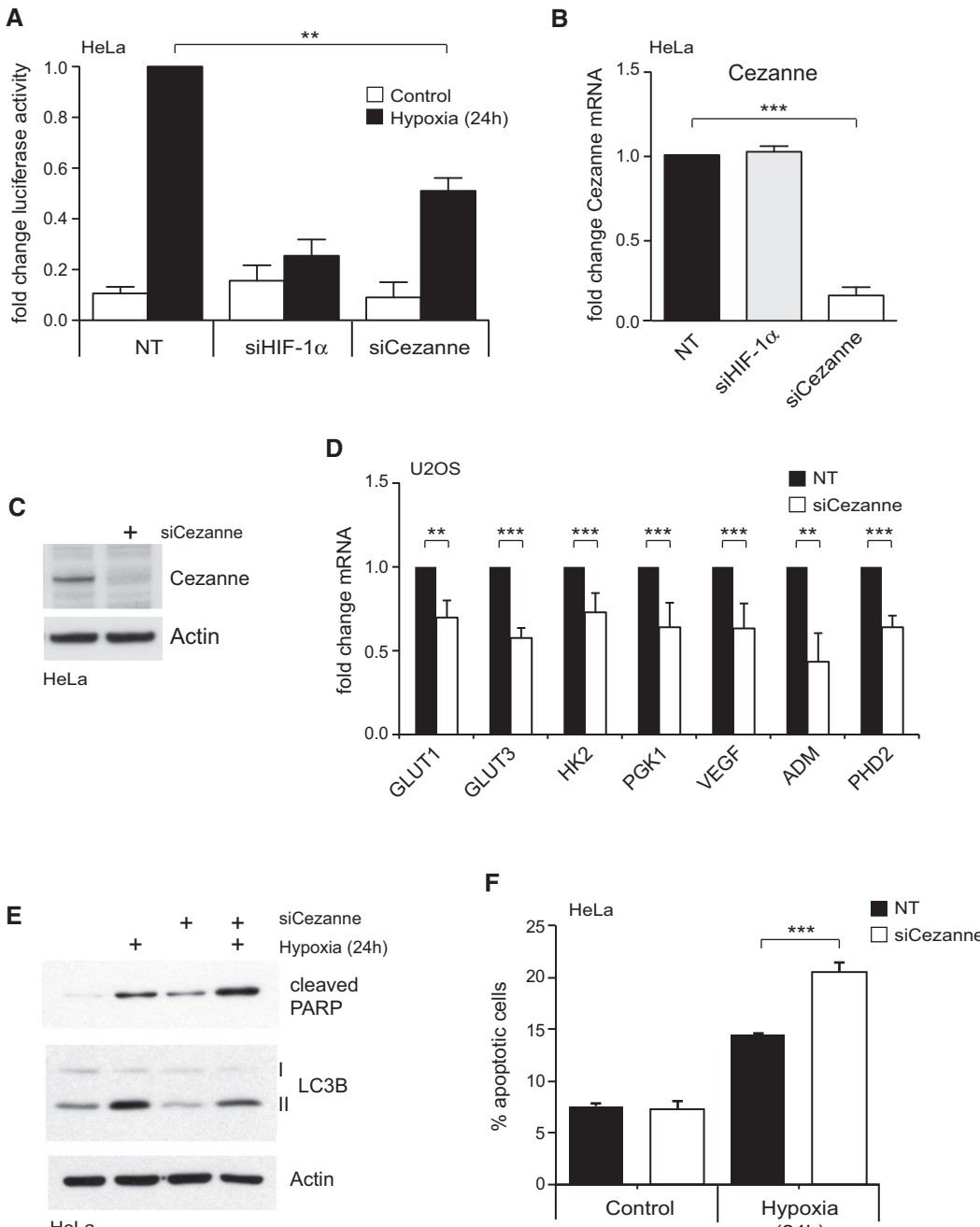

**Figure 1.  Cezanne is a regulator of HIF-1α transcriptional activity.**

A   Depletion of Cezanne decreased HRE-luciferase activity in hypoxia. Data were normalised to hypoxia-treated HeLa cells transfected with non-targeting (NT) siRNAs.

B, C  Knockdown of Cezanne significantly reduced Cezanne mRNA (B) and protein levels (C) relative to the NT control.

D   RT–PCR analysis of endogenous HIF-1α target genes in U2OS cells depleted of Cezanne and subjected to hypoxia. Data are shown as fold change when compared to NT control.

E   Cezanne-depleted cells show decreased LC3B-II and increased cleaved PARP levels.

F   Quantification of apoptotic cells using annexin V staining. Loss of Cezanne reduced cell viability in hypoxia.

Data information: All experiments were performed three times; bar graphs represent the mean plus standard deviation of these independent experiments. *P*-values were calculated using Student's *t*-test (\*\**P* < 0.01, \*\*\**P* < 0.001).

before [17], and also of p53 (Supplementary Fig S1F), suggesting that Cezanne can act as a positive and a negative regulator of transcription factors.

To determine whether knockdown of Cezanne derails adaptive responses to hypoxia, we examined expression of HIF target genes in Cezanne-depleted cells. HIF-1 controls a plethora of genes with

key functions in proliferation, energy metabolism, angiogenesis and apoptosis [18]. We designed a customised PCR screen to simultaneously analyse 81 HIF target genes that are involved in different physiological processes. These genes are expressed either ubiquitously or in a tissue-specific manner (Supplementary Tables S1 and S2; Supplementary Fig S2). In U2OS cells, mRNA levels of 23 out of the 81 studied genes were increased more than twofold in hypoxia (e.g. *CA9* (664-fold), *PHD3* (53-fold) or *ADM* (12-fold); Supplementary Fig S2A; Supplementary Table S2). Expression levels of 22 out of these 23 hypoxia-induced genes were reduced in Cezanne-depleted cells (Supplementary Fig S2A and B; Supplementary Table S2). The effect of Cezanne knockdown on several of the genes identified in the PCR array was confirmed by further analysis using different siRNA oligonucleotides (Fig 1D).

One of the main hypoxia-induced cellular responses is autophagy, a lysosomal-mediated degradation pathway used to survive starvation and stress. Amongst other functions, autophagy mediates resistance to chemotherapy in certain tumour cells by affecting their apoptotic potential [19]. We observed that the loss of Cezanne reduced the autophagy marker LC3B-II and led to PARP and Caspase-3 cleavage, indicative of apoptosis (Fig 1E, Supplementary Fig S1H). Annexin V staining showed that Cezanne knockdown had no apoptotic effect in normoxic cells, but in hypoxia resulted in approximately 25% more apoptotic cells (Fig 1F, Supplementary Fig S1I). This suggests that Cezanne-depleted cells, just like HIF-1-depleted cells, are sensitised for hypoxia-induced apoptosis [20,21].

## Cezanne controls HIF-1α protein levels

We next set out to determine how Cezanne controls HIF-1α transcriptional activity. It was recently reported that low oxygen levels induce Cezanne expression in cultured endothelial cells [22]. In contrast, Cezanne expression levels were not affected in hypoxic HeLa or U2OS cells (Fig 2, Supplementary Fig S3A), but importantly, loss of Cezanne caused decreased HIF-1α protein levels with or without hypoxia treatment in both the cytoplasm and the nucleus (Fig 2A). Depletion of Cezanne also reduced HIF-2α protein levels (Fig 2B), but did not alter expression levels of other transcription factors like Rb, β-catenin or the NF-κB subunit Rel A (Supplementary Fig S1G). qPCR analysis showed no significant change in HIF-1α mRNA levels upon Cezanne knockdown (Fig 2C), suggesting that Cezanne acts on HIF-1α protein at a post-translational level. Consistently, overexpression of GFP-tagged wild-type Cezanne increased HIF-1α protein levels, whereas expression of the inactive enzyme decreased HIF-1α levels (Fig 2D, Supplementary Fig S3B).

A defect in HIF-1α homeostasis was also observed in a Cezanne knockout mouse model. Primary mouse embryonic fibroblasts (MEFs) isolated from Cezanne knockout mice accumulated less HIF-1α under hypoxic conditions than cells from wild-type mice (Fig 2E). However, while HIF-1α deletion results in embryonic lethality [23,24], Cezanne knockout mice do not display an obvious phenotype except that male mice are infertile (http://www.phenogenomics.ca). Recently, it was shown that Cezanne knockout mice are more resistant to the intestinal bacterial pathogen *C. rodentium* [25]. It will be interesting to study how Cezanne knockout mice adapt to hypoxia.

Decreased HIF-1α protein upon Cezanne knockdown suggested a deregulation in upstream components of the HIF-1α degradation

machinery instead of transcriptional regulators. Consistently, Cezanne depletion did not change levels of FIH-1 and p300 [1,2] (Fig 2F). Surprisingly, proteins involved in HIF-1α degradation were also not affected in Cezanne-depleted cells (Fig 2F), indicating that Cezanne regulates HIF-1α degradation differently.

## HIF-1α is modified with Lys11-linked Ub chains

We had shown that the catalytic OTU domain of Cezanne is specific for Lys11-linked polyUb [26,27]. Also full-length Cezanne isolated from HEK293 cells is specific for Lys11-linked chains *in vitro* (Fig 3A). Moreover, using a Lys11 linkage-specific Ub antibody [28] (Supplementary Fig S3C), we found that overexpression of Cezanne resulted in a reduction of Lys11-linked Ub chains, while a catalytically inactive Cezanne mutant enriched this chain type in normoxic asynchronous cells (Fig 3B). Consistently, siRNA knockdown of Cezanne increased the amount of Lys11-linked polyUb in cells but had no obvious effect on Lys48 linkages (Fig 3C), which are more abundant in asynchronous cells. Together, this showed that Cezanne was able to regulate Lys11 polyubiquitination events in cells, raising the intriguing possibility that the observed effect of Cezanne on HIF-1α abundance was mediated by this chain type. Indeed, we detected endogenous Cezanne in the same complex as endogenous HIF-1α when it accumulated in cells under hypoxic conditions (Fig 3D). Importantly, when ubiquitinated HIF-1α was immunoprecipitated from cells treated with the proteasome inhibitor MG132, we identified Lys48 and Lys11 linkages in the precipitate by Ub linkage-specific antibodies (Fig 3E) and by Ub chain restriction analysis [27] (Supplementary Fig S3D). Knockdown of Cezanne increased the levels of Lys11 linkages on HIF-1α (Fig 3F).

Together, our results that the Lys11-specific DUB Cezanne affects hypoxia signalling suggest a role for atypical Lys11 linkages in this pathway. Lys11-linked polyUb has emerged as an independent Ub signal only recently, and important cellular roles, in particular in cell cycle regulation, have been suggested [29,30]. The only known Lys11 linkage-specific E2 enzyme, UBE2S, is cell cycle-regulated and associates with the E3 Ub ligase APC/C, promoting proteasomal degradation of mitotic regulators [31,32]. Our discovery of Lys11 linkages on HIF-1α adds another target for this chain type to the currently limited list of non-cell cycle-regulated proteins. UBE2S has also been suggested to mediate ubiquitination and degradation of pVHL, thus leading to increased HIF-1α levels [33]. We found that loss of Cezanne decreased HIF-1α levels independently of UBE2S (Supplementary Fig S3E). Consequently, what assembles Lys11 linkages in HIF-1α signalling has to be examined in more detail.

## Cezanne regulates non-proteasomal degradation of HIF-1α

We next studied the mechanism of HIF-1α degradation in Cezanne-depleted cells. An important regulator of protein turnover is the AAA$^+$ ATPase p97. Interestingly, the effect of Cezanne knockdown on HIF-1α levels in both normoxia and hypoxia could be rescued by co-depletion of p97 (Fig 4A). p97 regulates proteasomal degradation, ER-associated degradation as well as proteasome-independent processes such as autophagy and lysosomal sorting [34]. It was reported that Lys11-modified APC/C substrates are targeted for proteasomal degradation [31,32], but surprisingly, inhibition of the proteasome by MG132 or epoxomicin, which efficiently stabilised

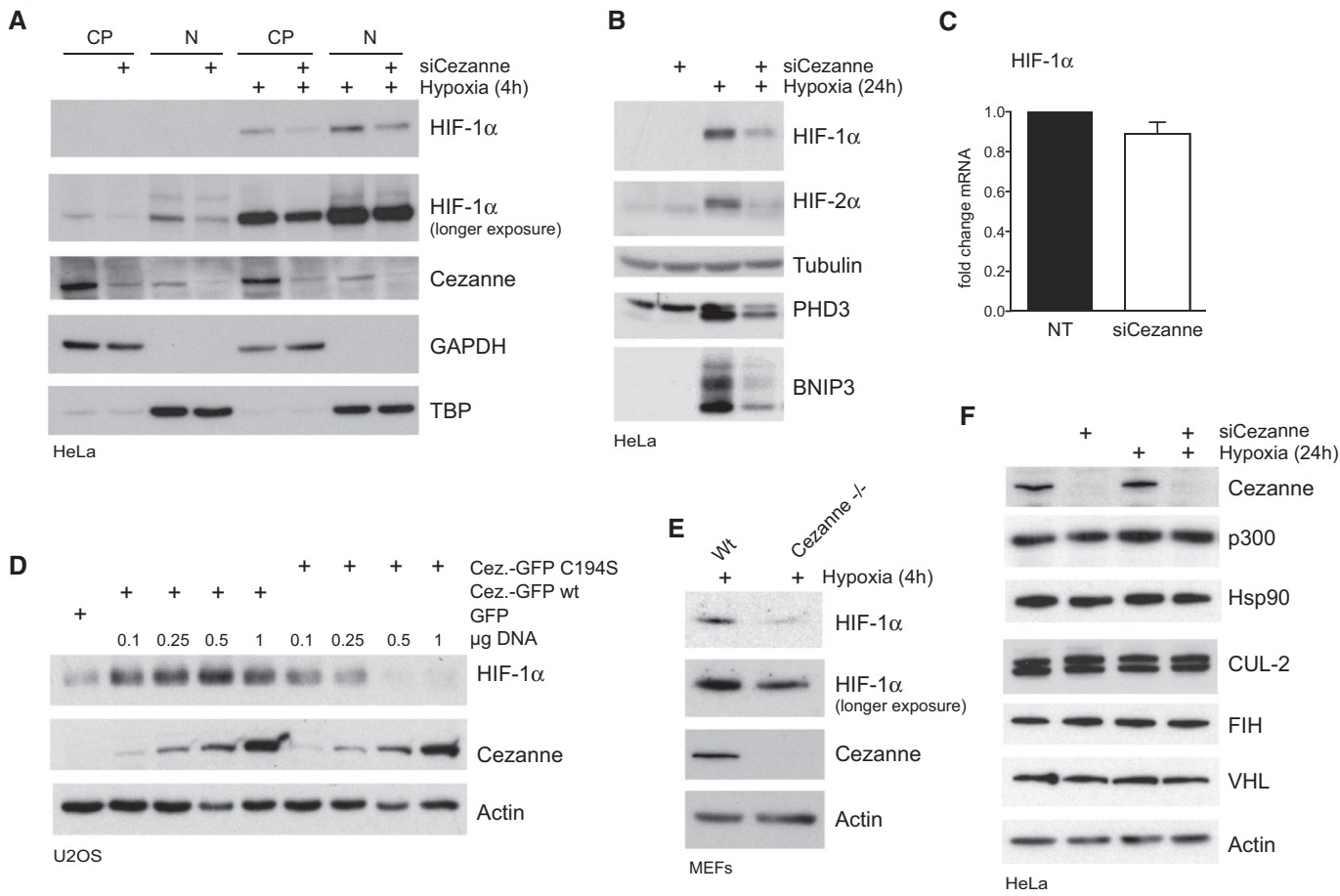

**Figure 2.  Cezanne is a positive regulator of HIF-1α protein levels.**

A   Subcellular fractionation revealed that loss of Cezanne caused reduced HIF-1α levels in cytoplasmic (CP) and nuclear (N) fractions in both untreated and hypoxia-treated cells.

B   Knockdown of Cezanne in HeLa cells subjected to hypoxia resulted in decreased HIF-1α and HIF-2α protein levels, and reduced hypoxia-induced PHD3 and BNIP3 levels compared to control cells.

C   RT–PCR analysis showed no significant changes of HIF-1α mRNA levels in Cezanne-depleted cells relative to control cells.

D   Overexpression of GFP-tagged wild-type (wt) or inactive (C194S) Cezanne in hypoxia-treated cells affected HIF-1α levels in a dose-dependent manner.

E   Mouse embryonic fibroblasts (MEFs) isolated from Cezanne knockout mice (Cezanne[−/−]) accumulated less HIF-1α in hypoxia than WT MEFs.

F   Depletion of Cezanne did not affect proteins involved in the regulation of HIF-1α activity or stability.

Data information: All experiments were performed three times; bar graphs represent the mean plus standard deviation of these independent experiments.

HIF-1α protein levels, did not rescue the effect of Cezanne knockdown on HIF-1α (Fig 4B, Supplementary Fig S4A and C). Cezanne depletion did not lead to accumulation of high molecular weight HIF-1α protein, but showed an apparently equal reduction in modified and unmodified HIF-1α species, also when cells were directly lysed in SDS sample buffer (Supplementary Fig S4B). In addition, co-depletion of Cezanne and the proteasome regulatory subunit RPN11 could not rescue reduced HIF-1α levels (Supplementary Fig S4D). Together, these data suggest that Cezanne does not impact solely on proteasomal degradation of HIF-1α, and raise the general question whether the canonical HIF-1α degradation machinery is required for Cezanne-mediated HIF-1α downregulation. To test this, we inhibited the PHD enzymes, the most upstream signal required to trigger CRL2[VHL] ubiquitination. Suppression of PHD activity by the hydroxylase inhibitor DMOG

averts HIF-1α hydroxylation and stabilises HIF-1α (Fig 4C, compare lane 1 and 3), but protein levels of HIF-1α and its transcriptional activity were still decreased in Cezanne-depleted cells (Fig 4C and D). Interestingly, in the complete absence of functional pVHL as in the renal cell carcinoma cell lines RCC4 and A498, the effect of Cezanne knockdown was rescued (Fig 4E and F). Consistently, RCC4 cells expressing HA-tagged pVHL showed again reduced HIF-1α levels upon Cezanne knockdown (Fig 4E). This suggests that while Cezanne does not affect pVHL levels (Fig 2F), it regulates HIF-1α homeostasis in a pVHL-dependent way. HIF-1α hydroxylation is a prerequisite for pVHL-mediated HIF-1α degradation via the proteasome. The apparent dependence of Cezanne on pVHL, but not hydroxylation, is unexplained. Attempts to test the action of Cezanne on prolyl mutant HIF-1α have so far been unsuccessful.

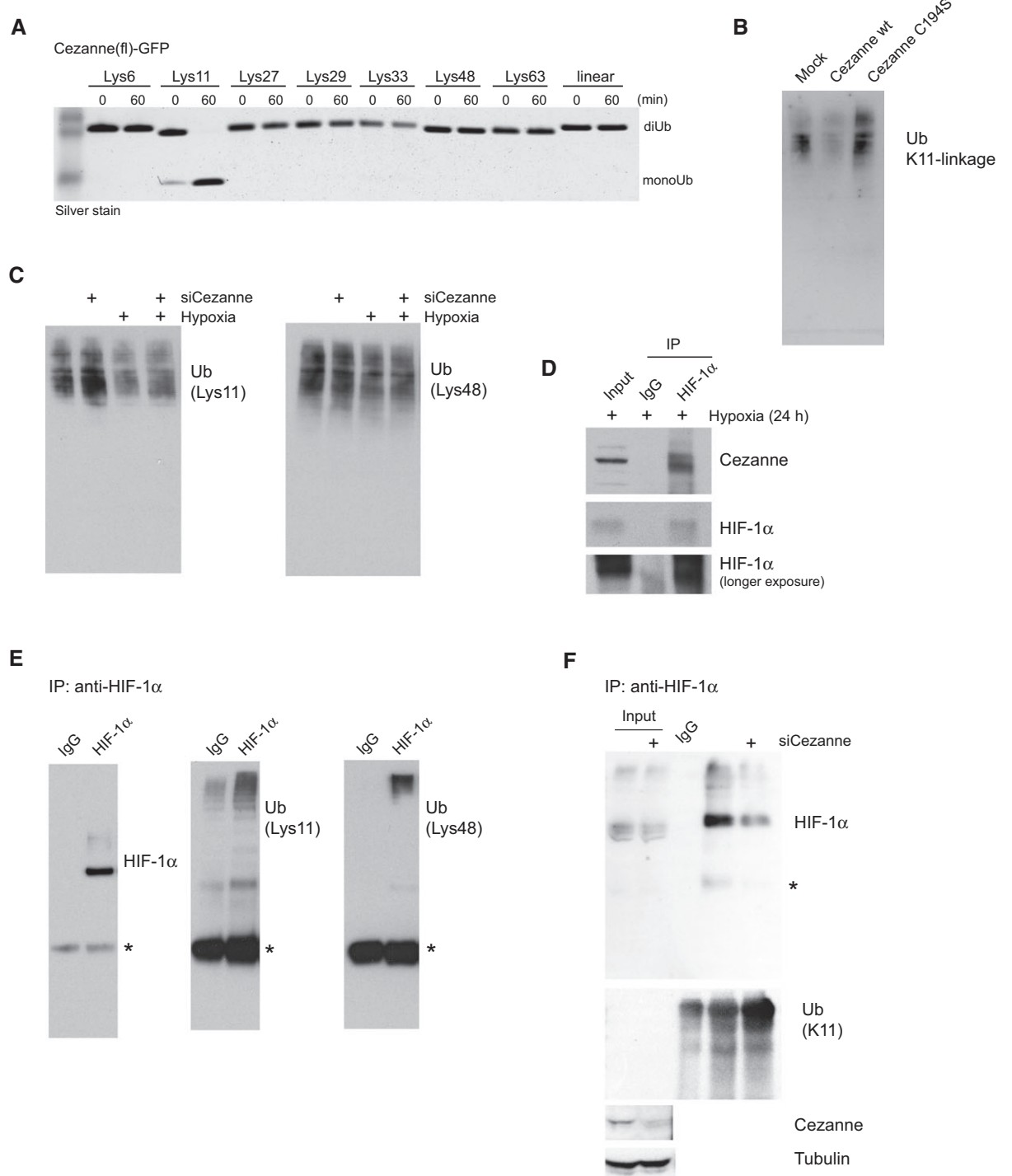

**Figure 3.  Lys11-linked polyUb in hypoxia.**

A    Full-length (fl) GFP-tagged Cezanne isolated from HEK293 cells specifically cleaved Lys11-linked diUb in an *in vitro* DUB assay comprising eight differently linked Ub dimers.

B    Global levels of ubiquitin Lys11 linkages in total cell extracts were determined using a linkage-specific antibody (see Supplementary Fig S3C). Overexpression of WT Cezanne decreased the amount of Ub chains containing Lys11 linkages, whereas a catalytically inactive Cezanne (C194S) enriched this linkage type.

C    Knockdown of Cezanne caused increased levels of Lys11-linked polyUb relative to the NT control, but did not affect K48 linkages.

D    Cezanne and HIF-1α co-immunoprecipitated in hypoxia-treated cells.

E, F  Polyubiquitinated HIF-1α was immunoprecipitated with HIF-1α antibody from MG132-treated HeLa cells. Ub linkage-specific antibodies detected Lys11 and Lys48 linkages in the high-molecular-weight smear enriched with HIF-1α antibody, showing that HIF-1α is modified with Lys11-linked polyUb (E). Knockdown of Cezanne increased Lys11 linkages on HIF-1α (F). *, heavy chain.

Data information: All experiments were performed three times.

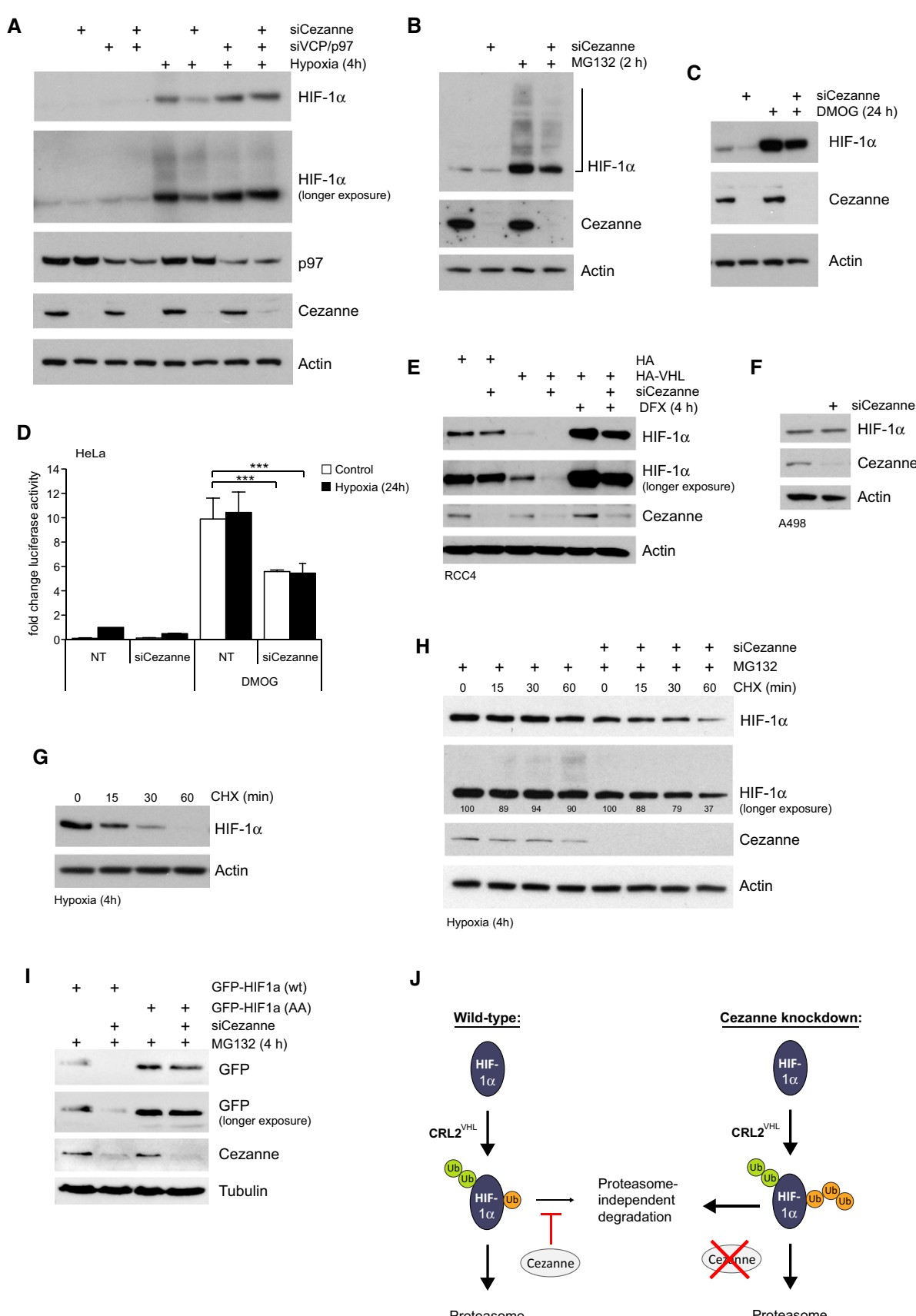

**Figure 4.**

HIF-1α has a very high turnover rate in normoxia with a half-life of only 5–8 min [35]. Lack of oxygen inhibits PHD activity [36], which increases HIF-1α levels and transcriptional activity. HIF-1α-mediated transcription of PHD2 and PHD3 establishes a negative feedback mechanism [37–39], which leads to partial destabilisation of HIF-1α, even in hypoxia. To investigate HIF-1α turnover under hypoxic conditions, cells were subjected to 4 h hypoxia before protein translation was blocked using cycloheximide (Fig 4G). As expected, HIF-1α turnover was significantly delayed in hypoxia, but protein levels still declined after 60 min, highlighting that HIF-1α is also degraded in hypoxia. As previously shown [40], this can be rescued by proteasome inhibition (Fig 4H, lanes 1–4), and ubiquitinated HIF-1α species start to accumulate at later time points (Fig 4H, see longer exposure). Importantly, adding MG132 failed to rescue HIF-1α degradation in Cezanne-depleted cells in this experiment (Fig 4H, lanes 5–8). Overall, these results imply that loss of Cezanne promotes degradation of HIF-1α in a proteasome-independent manner.

Two recent reports revealed that HIF-1α is targeted for lysosomal degradation via chaperone-mediated autophagy (CMA) [9,41]. Since decreased HIF-1α levels in Cezanne-depleted cells could only partially be rescued by combined inhibition of proteasome and lysosome activity (Supplementary Fig S4E), we interfered with the KFERQ-like motif in HIF-1α [41], which has been identified in all CMA substrates. Interestingly, mutation of this motif in GFP-tagged HIF-1α generated a protein that was no longer affected by Cezanne knockdown (Fig 4I). This suggests that Cezanne regulates the CMA-mediated degradation of HIF-1α.

In summary, we identified Cezanne as a new regulator of HIF-1α homeostasis. Our data show that Cezanne controls HIF-1α transcriptional activity by preventing proteasome-independent degradation of HIF-1α. Loss of Cezanne may alter the ubiquitination pattern of HIF-1α, thereby causing excessive clearance of the transcription factor. Even though this process is not reliant on full hydroxylase activity, it depends on pVHL as demonstrated in pVHL-deficient cell lines.

Cezanne and Cezanne-2 are the only Lys11 linkage-specific DUBs described to date [27]. Cezanne is not cell cycle-regulated (www.cyclebase.org) and its depletion does not readily change cell cycle distribution (Supplementary Fig S4G), suggesting that it controls other Lys11-mediated processes in cells. A recent paper proposed a role for Cezanne in EGF receptor degradation [42], but the role of Lys11 linkages in this process was not studied. Interestingly, these authors also reported that Cezanne is overexpressed in a large percentage of breast cancers and that high expression levels of Cezanne correlated with poor patient prognosis. Our functional analysis of the effects of Cezanne on cell viability indicates that indeed, Cezanne would support a pro-survival response in hypoxia. These data, together with our finding that Cezanne is required for HIF-1α homeostasis, make the enzyme an important target for future studies, and potentially for pharmacological intervention.

# Materials and Methods

### Cell culture and transfection

Human cell lines (U2OS, HeLa, HEK293, RCC4, A498) and mouse embryonic fibroblasts were cultured in Dulbecco's modified Eagle medium (Life Technologies) supplemented with 10% (v/v) foetal bovine serum (Thermo Scientific), 50 U/ml penicillin and 50 μg/ml streptomycin (GE Healthcare) at 37°C and 5% $CO_2$. A498 cells were a kind gift from R. Pawlowski, Institute for Molecular Health Sciences, ETH Zurich, Switzerland. U2OS-HRE and HeLa-HRE-luciferase cells were described in [43]. Hypoxia treatment at 1% $O_2$ was achieved using an INVIVO$_2$ hypoxia workstation (Ruskinn) or Don Whitley H35 workstation. To avoid reoxygenation, cells were lysed in the workstation. For immunoblot and real-time RT–PCR, $2 \times 10^5$ cells were seeded in 6-well plates and transfected after 24 h with either 30 nM siRNA duplexes using INTERFERin transfection reagent (Polyplus) or 1 μg plasmid DNA using GeneJuice (Merck Biosciences). Further information on siRNA oligonucleotides can be found in the Supplementary Information.

### Luciferase reporter assay

$2 \times 10^4$ cells were seeded in 24-well plates and transfected with 40 nM siRNA duplexes using INTERFERin transfection reagent (Polyplus). A PG13-luciferase construct [44] was co-transfected with siRNA using jetPRIME transfection reagent (Polyplus). Forty-eight hours post-transfection cells were lysed in 1× passive lysis buffer (Promega), and luciferase assays were performed according to the

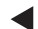

**Figure 4. Cezanne regulates proteasome-independent degradation of HIF-1α.**

A    Decreased HIF-1α levels in Cezanne knockdown cells were complemented by co-depletion of p97.

B, C    Reduced HIF-1α levels caused by Cezanne depletion can neither be complemented with the proteasome inhibitor MG132 (B) nor the hydroxylase inhibitor DMOG (C).

D    Control and Cezanne-depleted cells were treated with or without hypoxia and DMOG. Loss of Cezanne reduced HIF-1α-dependent luciferase activity also when hydroxylases were inhibited.

E, F    Knockdown of Cezanne in VHL-deficient RCC4 cells (E, lanes 1 and 2) and A498 cells (F) had no effect on HIF-1α levels. Expression of exogenous VHL sensitised RCC4 cells for Cezanne regulation (E, lanes 3–6).

G    Cells were exposed to hypoxia and subsequently treated with cycloheximide to block protein translation. HIF-1α degradation under these conditions was followed over time.

H    Cells were treated with hypoxia before cycloheximide and MG132 were added. HIF-1α degradation could be rescued with proteasome inhibition in control cells but not in Cezanne-depleted cells.

I    Loss of Cezanne did not affect protein levels of HIF-1α with mutations in the KFERQ-like motif (GFP-HIF-1α (AA)).

J    Model: Cezanne prevents proteasome-independent HIF-1α degradation.

Data information: All experiments were performed three times, experiment in panel (E) was repeated four times; bar graphs represent the mean plus standard deviation of these independent experiments. *P*-values were calculated using Student's *t*-test (***$P < 0.001$).

manufacturer's instructions (Luciferase Assay System, Promega). U2OS- and HeLa-HRE-luciferase cells were cultured in 1% $O_2$ prior to lysis. U2OS-κB-luciferase cells [45] were treated with 10 ng/ml TNF-α (Peprotech) for 6 h prior to harvest. Results were normalised for protein concentration with all experiments being performed a minimum of three times before calculating means and standard deviations as shown in the figures.

### Analysis of gene expression levels by real-time RT–PCR

DNase-treated total RNA was reverse-transcribed using QuantiTect Reverse Transcription Kit (Qiagen). RT–PCR was performed on a Rotor-Gene 6000 (Qiagen) using QuantiFast SYBR Green PCR Kit (Qiagen). RT–PCR data were analysed using the comparative concentration module of the Rotor-Gene software, which is based on [46]. Signal for the gene of interest was normalised to signal for *ACTB*, and then fold change was calculated relative to calibrator sample. For each primer pair, the formation of a single product was confirmed by melt curve analysis [47].

Custom PCR Arrays: cDNA was made using $RT^2$ First Strand Kit (Qiagen). RT–PCR was performed using $2\times RT^2$ SYBR Green ROX FAST Mastermix (Qiagen). Threshold cycle ($C_T$) for each well was calculated using the Rotor-Gene software after threshold was manually defined. Subsequently, fold change was calculated by the web-based PCR Array Data Analysis Tool provided by Qiagen/SABiosciences.

### Immunoblot

Cells were lysed in 50 mM Tris–HCl (pH 7.4), 150 mM NaCl, 5 mM $MgCl_2$, 1% (v/v) IGEPAL CA-630, 0.5% (v/v) sodium deoxycholate, 0.1% (v/v) SDS, 100 mM NaF, 5 mM N-ethylmaleimide, 10 mM iodoacetamide, 1 tablet/10 ml cOmplete, Mini, EDTA-free protease inhibitors (Roche) and 2 μl/ml benzonase nuclease (≥ 250 units/μl). Cells were treated with 20 μM MG132 (Sigma), 2 μM epoxomicin (Enzo Life Sciences), 200 μM deferoxamine (Sigma), or 50 μM chloroquine (Sigma) as specified in the main text. SDS–PAGE and immunoblots were carried out using standard protocols. A list of antibodies used can be found in the Supplementary Information.

### Apoptosis assay

Apoptosis was measured by Annexin V staining using the Nexin assay kit (Millipore). Cells were processed and stained according to the manufacturer's instructions and acquired using flow cytometry (Guava EasyCyte, Millipore).

### Immunoprecipitation

For precipitation of endogenous HIF-1α, Cezanne-depleted and control cells were treated with 20 μM MG132 for 2 h (6-well plate) and subsequently lysed in 100 μl lysis buffer per well (50 mM Tris–HCl (pH 7.5), 150 mM NaCl, 1% (v/v) IGEPAL CA-630, 2 mM EDTA, 1 mM dithiothreitol, 100 mM NaF and 1 tablet/10 ml cOmplete, Mini, EDTA-free protease inhibitors (Roche)). Cleared cell lysate was rotated at 4°C for 3 h with 2 μg of anti-HIF-1α antibody (sc-10790, Santa Cruz) and Protein A Sepharose (GE Healthcare), which was added after 2 h. Immobilised antigene–antibody

complex was then washed three times with PBS and eluted in 2× LDS sample buffer (Invitrogen).

**Supplementary information** for this article is available online: http://embor.embopress.org

### Acknowledgements

We thank M. Pasparakis for providing us with Cezanne$^{-/-}$ MEFs, E. Chilvers and B. Brüne for allocating hypoxia workstations, T.E.T. Mevissen, L. James, F. Hauler, F. Randow and I. Dikic for reagents and technical advice. We thank members of the Komander, the Rocha and the Bremm laboratory for helpful discussions. A.B. was supported by the Medical Research Council and the German Research Foundation (DFG) and is currently funded by the DFG (Emmy Noether Programme). S.R. is funded by a Senior Research Fellowship from Cancer Research UK and by the Wellcome Trust (097945/Z/11/Z). Work in the D.K. laboratory is supported by the Medical Research Council [U105192732], the European Research Council [309756], the Lister Institute for Preventive Medicine and the EMBO Young Investigator Program.

### Author contributions

AB, SM, JM and SR performed experiments; AB, SR and DK conceived the project and wrote the manuscript.

### Conflict of interest

DK is a part of the DUB Alliance that includes Cancer Research Technology and FORMA Therapeutics. The remaining authors declare that they have no conflict of interest.

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
