## [Review Process File · EMBO Reports]

Manuscript EMBO-2014-38850

Cezanne (OTUD7B) regulates HIF-1 α homeostasis in a proteasome-independent manner

Sonia Moniz, Julia Mader, Sonia Rocha, David Komander and Anja Bremm

Corresponding authors: Anja Bremm, Buchmann Institute for Molecular Life Sciences; Sonia Rocha, College of Life Sciences, Centre for Gene Regulation and Expression, University of Dundee; David Komander, Medical Research Council Laboratory of Molecular Biology

Review timeline:

Submission date:	30 March 2014
Editorial Decision:	28 April 2014
Revision received:	29 August 2014
Editorial Decision:	15 September 2014
Revision received:	24 September 2014
Accepted:	29 September 2014

Transaction Report:

Editor: Nonia Pariente

1st Editorial Decision

28 April 2014

Thank you for your submission to EMBO reports. We have now received reports from the three referees that were asked to evaluate your study, which can be found at the end of this email. As you will see, although all referees find the topic of interest, they all raise numerous concerns and ultimately consider that the study is insufficiently conclusive as it stands.

As the reports are below, I will not detail them here. However, it is clear that substantial additional characterization of the role of Cezanne and K11-linked ubiquitin chains in HIF-1 regulation is needed, such as more conclusively analyzing the proteasomeal and hydroxylation independence, the specificity of Cezanne, showing that it indeed removes K11-linked polyubiquitin from HIF-1 α , as well as strengthening the data throughout the manuscript. In general, all experiments should be performed at least three independent times, and quantification and statistical analyses provided, especially as some of the effects seem modest.

Given the constructive comments from the referees, we would be happy to invite revision of your study. Please note that it is our policy to undergo one round of revision only and thus, acceptance of your study will depend on the outcome of the next, final round of peer-review. All referee concerns seem reasonable and should be addressed.

Revised manuscripts must be submitted within three months of a request for revision unless previously discussed with the editor; they will otherwise be treated as new submissions. Revised manuscript length must be a maximum of 30,000 characters (including spaces). When submitting your revised manuscript, please also include editable TIFF or EPS-formatted figure files, a separate PDF file of any Supplementary information (in its final format) and a letter detailing your responses to the referees.

Please get in touch with me if I can be of any help during the revision process.

Referee #1:

Recent work suggested that the E2 Ube2S (E2-EPF) is involved in regulation of VHL, a ubiquitin ligase that is responsible for the degradation of the Hif1 α -transcription factor. Other work had shown that Ube2S assembles an atypical K11-linked ubiquitin chain. These findings suggested that atypical ubiquitin chains might play a role in regulating the cellular response to hypoxic conditions. In this paper, Bremm et al. build on this notion and identify the deubiquitylating enzyme Cezanne as a potential regulator of Hif1 α -activity. Cezanne is a K11-specific DUB, confirming a role of this ubiquitylation topology for Hif1 α -regulation. Further experiments suggested that Cezanne regulates Hif1 α -levels by counteracting an ill-defined proteolytic pathway that does not appear to depend on the proteasome (contrary to what is thought to be the function of K11-linked ubiquitin chains).

This paper contains interesting findings and many experiments have been performed and interpreted well. A function of Cezanne in regulating Hif1 α -stability would be an interesting result, as would be a potential role of k11-chains in processes other than proteasomal degradation. However, in its current state, the manuscript does not fully support these conclusions, as outlined below.

Major issues:

1. I am concerned about the authors' conclusion that the proteasome is not involved in the regulation of Hif1 α -levels by Cezanne. In their experiments, they usually treat cells with siRNAs against Cezanne for 48h (the methods section is not very detailed, so I might have been wrong; in any case, it would be more than 24h), yet proteasome inhibitors are added for only 2h. It is often the case that a phenotype is established fairly early after siRNA-transfection, and cells become less responsive to many treatments once the phenotype has been established. Thus, to strengthen their argument, the authors should (1) use siRNA-dependent proteasome inhibition (Rpn11 works well) to show that loss of proteasome activity does not rescue loss of Cezanne (acknowledging that it will also affect other processes, but proteasome and Cezanne inhibition will occur on the same time scale) and (2) use additional, more potent proteasome inhibitors (Velcade) for longer times.

2. The analysis of Cezanne-depletion is missing an important control: if the author's hypothesis is correct, then loss of Cezanne should neither affect other transcription factors nor the general transcription machinery. The authors should either run a microarray in Cezanne-depleted vs control-depleted cells (this can easily be outsourced and should be straightforward, and it would be very unbiased) or at the minimum, they should control for effects of Cezanne-depletion on other transcription factors with well established reporter systems (NF κ B, beta-catenin etc).

3. Fig. 1a: the screen data is also missing important information: what is the mean luciferase activity over all assays and where is a 2-fold standard deviation from this mean? This would be important information for judging whether the relatively weak effects of Cezanne depletion are statistically significant.

Minor issues:

1. Fig. 1H: the cleaved caspase blot needs to be replaced (low quality); also, a decrease in LC3B is seen upon Cezanne-depletion even without hypoxia, pointing to some effects of Cezanne depletion on a general mechanism such as autophagy (this provides another argument for the microarray,

which would give a complete picture of the cell's transcriptional state after Cezanne depletion, potentially revealing stress responses etc).

2. Fig. 2F: it is not really surprising that loss of Cezanne does not lead to a change in Hsp90 - it would be more interesting to look at substrate adaptors of this chaperone. This should be reworded.

3. Fig. 3C: to this reviewer, the depletion of Cezanne has weak (significant?) effects on K11-linkages. Also, it is an unfair comparison between K11- and K48-chains, as K48-linked chains appear to be much more prominent in asynchronous cells than K11-linked chains (see Junmin Peng's mass spec or Dixit's antibody, for example). Please discuss this more carefully.

4. Fig. 3F: needs to be repeated, not convincing at this point.

Referee #2:

This paper describes a role for the deubiquitylating enzyme Cezanne, in inhibiting the degradation of HIF-1 α and probably HIF-2 α . It is proposed that this occurs by removal of ubiquitin lysine 11 linkages, which otherwise promote destruction of HIF-1 α by a proteasome independent pathway. It is also proposed that this process is dependent on VHL but independent of hydroxylation.

If true this would be an interesting finding revealing further complexity in pathways that regulate HIF-1 α by degradation.

Points to be considered or clarified:

(i) The finding that the pathway is dependent on VHL but independent of hydroxylation is odd since known VHL functions on the HIF pathway are hydroxylation dependent. The data presented in support of this statement is not altogether convincing. The data presented in lanes 1 and 2 of figure 4C is critical. From the illustration provided it appears possible that there is some reduction in the HIF-1 α signal. How many times was this experiment repeated? Careful repetition is required to be sure of this point. Since this is an important (and unexpected) aspect of the report the authors should perform additional experiments. Intervention in other RCC cells, effects on HIF-2 α and effects on transfected prolyl mutated forms of HIF-1/2 α (which should not be recognized by VHL) should all be readily possible and should be capable of confirming or refuting this interpretation.

(ii) Figure 3F is not entirely convincing. How many times was this experiment repeated?

(iii) The authors should consider intervening on UBE2S and measuring the effect on HIF-1 α stability to reinforce their conclusions.

Referee #3:

Cezanne (also known as OTU7B) is a deubiquitinating enzyme shown in previous work to disassemble K11-linked polyubiquitin preferentially. This submission from Bremm et al. reports the very interesting observation that Cezanne contributes to the regulation of HIF-1 α , an important transcription factor. Data are provided that show that loss of Cezanne destabilizes HIF-1 α , and that, surprisingly, the HIF-1 α degradation that results is proteasome independent. These all are strengths of the paper. However, there are several problems. As elaborated below, some of the paper's claims are only weakly supported, the mechanism that links Cezanne to HIF-1 α - a key aspect of the study's significance - was investigated rather superficially, and the quality of some of the experimental data is disappointing.

1. The claim that p97 is involved in HIF-1 α degradation (see Abstract and pages 4 & 12) is based on very superficial observations. Because p97 has so many diverse functions (e.g., membrane fusion & trafficking, gene expression, and both proteasomal and lysosomal degradation pathways), the effects of p97 knockdown on HIF-1 α levels as described by Bremm et al. could be very indirect. Without

evidence from more detailed studies, there is no reason to think that p97 is involved directly in HIF-1 α degradation.

2. Despite providing evidence that Cezanne disassembles K11-polyubiquitin, that Cezanne and HIF-1 α can co-IP, and that immunoprecipitated HIF-1 α contains K11-linked polyubiquitin, the authors never showed that Cezanne specifically removes K11-linked polyubiquitin from HIF-1 α conjugates (see #3, below). They certainly imply that Cezanne has that function, although their model (Fig 4G) is rather vague on this point. This is a key issue raised by the authors' observations and, as the tools appear to be available to investigate it, should be examined experimentally.

3. (Fig. 3F) The extent of deubiquitination by Cezanne or OTUB1 of HIF-1 α conjugates is difficult to gauge from the western blot shown. The Cezanne digestion in particular appears to have had a modest effect. Presumably, detection used an anti-ubiquitin antibody. The experiment would be much more informative if the blots were also developed with anti-K11 and anti-K48 antibodies (or, even better, if the different linkages were quantified by mass spectrometry). Is HIF-1 α deubiquitinated completely by the combination of Cezanne and OTUB1?

4. The authors need to show more clearly the relationship between Cezanne and USP20 with respect to HIF-1 α stabilization. Are the effects of combining siRNA knockdowns of the two DUBs additive or synergistic? Can USP20 facilitate HIF-1 α deubiquitination by Cezanne?

5. In Fig. 1A, setting the threshold at 75% seems rather arbitrary but, for the authors' purpose, reasonable. However, the lack of error bars is disturbing. Without appropriate statistics (e.g., standard errors determined from multiple experiments), it's impossible to gauge the reliability of these measurements.

6. Fig. 3A essentially repeats the authors' published results (e.g., Fig 2 in ref 26) and should be removed.

7. (p. 8) I disagree with the authors' contention that "...less Lys11-linked ubiquitin chains in hypoxia-treated cells compared to control cells (Fig 3C) [suggests]...that this chain type is differentially regulated by an external stimulus". An alternative explanation is that an external stimulus could limit ubiquitin availability, which in turn could affect availability of different E2-ubiquitin thioester species.

8. (p. 12, 1st para) The sentence "Our data suggest that depletion of p97..." isn't clear

1st Revision - authors' response

29 August 2014

Point-by-point response to the reviewer's comments:

First, we would like to thank all three reviewers for their time and the constructive comments, which we were able to address as specified below. Due to these insights and suggestions we could improve our manuscript substantially. In total we have included 10 additional figures into the main report and the *Expanded View* section.

The most important improvements in our revised manuscript are:

- **Fig. E4C-D** supporting that Cezanne regulates HIF-1 α in a proteasome-independent manner
- **Fig. 4I** showing that protein levels of chaperone-mediated autophagy (CMA)-deficient HIF-1 α are not decreased in Cezanne-depleted cells anymore, which suggests a role for Cezanne in HIF-1 α degradation by CMA
- **Fig. 4F & E4E** further supporting that Cezanne's effect on HIF-1 α depends on pVHL
- **Fig. 3F** demonstrating that Cezanne can regulate Lys11 linkages in HIF-1 α

immunoprecipitates

Referee #1:

Recent work suggested that the E2 Ube2S (E2-EPF) is involved in regulation of VHL, a ubiquitin ligase that is responsible for the degradation of the Hif1 α -transcription factor. Other work had shown that Ube2S assembles an atypical K11-linked ubiquitin chain. These findings suggested that atypical ubiquitin chains might play a role in regulating the cellular response to hypoxic conditions. In this paper, Bremm et al. build on this notion and identify the deubiquitylating enzyme Cezanne as a potential regulator of Hif1 α -activity. Cezanne is a K11-specific DUB, confirming a role of this ubiquitylation topology for Hif1 α -regulation. Further experiments suggested that Cezanne regulates Hif1 α -levels by counteracting an ill-defined proteolytic pathway that does not appear to depend on the proteasome (contrary to what is thought to be the function of K11-linked ubiquitin chains).

This paper contains interesting findings and many experiments have been performed and interpreted well. A function of Cezanne in regulating Hif1 α -stability would be an interesting result, as would be a potential role of k11-chains in processes other than proteasomal degradation. However, in its current state, the manuscript does not fully support these conclusions, as outlined below.

Major issues:

1. I am concerned about the authors' conclusion that the proteasome is not involved in the regulation of Hif1 α -levels by Cezanne. In their experiments, they usually treat cells with siRNAs against Cezanne for 48h (the methods section is not very detailed, so I might have been wrong; in any case, it would be more than 24h), yet proteasome inhibitors are added for only 2h. It is often the case that a phenotype is established fairly early after siRNA-transfection, and cells become less responsive to many treatments once the phenotype has been established. Thus, to strengthen their argument, the authors should (1) use siRNA-dependent proteasome inhibition (Rpn11 works well) to show that loss of proteasome activity does not rescue loss of Cezanne (acknowledging that it will also affect other processes, but proteasome and Cezanne inhibition will occur on the same time scale) and (2) use additional, more potent proteasome inhibitors (Velcade) for longer times.

We agree with reviewer #1 that the time scales of siRNA knockdown of Cezanne and the pharmacological inhibition of the proteasome deviate. To strengthen our data obtained in MG132-treated cells, we followed reviewer #1's suggestion and co-depleted the proteasome regulatory subunit Rpn11 and Cezanne by siRNA knockdown. Fig. E4D in our revised manuscript shows that under these conditions, HIF-1 α protein levels are still reduced as compared to control cells when Cezanne is co-depleted.

Although MG132 treatment for 2 h appears to efficiently stabilize HIF-1 α protein (Fig. 4B, compare lane 1 & 3), we now also used the irreversible proteasome inhibitor epoxomicin. Cells were subjected to epoxomicin for 4-6 h before lysis. As demonstrated in Fig. E4C, Cezanne knockdown resulted in decreased HIF-1 α protein levels in this experimental setup as well. Taken together, our new data further support the hypothesis that Cezanne regulates HIF-1 α homeostasis in a proteasome-independent way. (Interestingly, our new Fig. 4I suggests that regulation may occur via chaperone-mediated autophagy).

2. The analysis of Cezanne-depletion is missing an important control: if the author's hypothesis is correct, then loss of Cezanne should neither affect other transcription factors nor the general transcription machinery. The authors should either run a microarray in Cezanne-depleted vs control-depleted cells (this can easily be outsourced and should be straightforward, and it would be very unbiased) or at the minimum, they should control for effects of Cezanne-depletion on other transcription factors with well established reporter systems (NF κ B, beta-catenin etc).

To address reviewer #1's concern that the effect of Cezanne knockdown on HIF-1 α is due to a general decrease of transcriptional activity in the cell, we compared activity of NF- κ B and p53 using luciferase-based reporter systems in control and Cezanne-depleted cells. As already shown by Enesa

et al. (2008) (and in contrast to HIF-1 α), NF- κ B-dependent reporter gene activity was upregulated in Cezanne-depleted cells (Fig. E1E). Interestingly, we observed that transcriptional activity of p53 was also increased (Fig. E1F), suggesting that Cezanne has differential effects on these transcription factors and can act as a positive and negative regulator.

In the light of these results, and the scope of this manuscript we hope that reviewer #1 can accept our decision of not performing a microarray analysis in this study, which we could not feasibly perform in the time-scale of this revision as it is not an established technique in our laboratories.

3. Fig. 1a: the screen data is also missing important information: what is the mean luciferase activity over all assays and where is a 2-fold standard deviation from this mean? This would be important information for judging whether the relatively weak effects of Cezanne depletion are statistically significant.

We agree that this experiment was not performed optimally, which is partly due to the design of the screen. The presented screen was conducted in triplicate with very good Z-score (>0.8). In the revised version, we placed less emphasis on it, since we merely used it as a starting point for our analyses of Cezanne's role in HIF-1 α regulation and we do not conclude anything further based on it. We have moved this figure into the Expanded View section (Fig. E1A).

Although we cannot provide more details for the above-mentioned figure, our manuscript contains all necessary experiments and controls to confirm the effect of Cezanne knockdown on HIF-1 α transcriptional activity (approx. 40% decrease in activity): sufficient experimental repeats, various siRNA oligonucleotides, and different cell lines (Fig. 1A, 1D, E1B, E1D, E2). As the reviewer implies, these experiments go beyond the qualitative findings in the screen, and are more solid and quantitative.

Minor issues:

1. Fig. 1H: the cleaved caspase blot needs to be replaced (low quality); also, a decrease in LC3B is seen upon Cezanne-depletion even without hypoxia, pointing to some effects of Cezanne depletion on a general mechanism such as autophagy (this provides another argument for the microarray, which would give a complete picture of the cell's transcriptional state after Cezanne depletion, potentially revealing stress responses etc).

The cleaved-caspase 3 blot was removed from Fig. 1E, and a blot of better quality can be found in Fig. E1H.

2. Fig. 2F: it is not really surprising that loss of Cezanne does not lead to a change in Hsp90 - it would be more interesting to look at substrate adaptors of this chaperone. This should be reworded.

We have reworded this paragraph.

3. Fig. 3C: to this reviewer, the depletion of Cezanne has weak (significant?) effects on K11-linkages. Also, it is an unfair comparison between K11- and K48-chains, as K48-linked chains appear to be much more prominent in asynchronous cells than K11-linked chains (see Junmin Peng's mass spec or Dixit's antibody, for example). Please discuss this more carefully.

We discussed Fig. 3C more carefully as suggested.

4. Fig. 3F: needs to be repeated, not convincing at this point.

The recently established method of ubiquitin chain restriction analysis (Mevisen *et al.*, 2013) was shown to be efficient on different *in vitro* ubiquitylated model substrates. In chase of polyubiquitylated HIF-1 α immunoprecipitated from HeLa cells (treated with MG132) we observed complete removal of ubiquitin conjugates by USP21, which shows no preference for a certain

linkage type.

However, incubation of polyubiquitylated HIF-1 α with Lys11 linkage-specific Cezanne or Lys48 linkage-specific OTUB1 only partially collapsed the attached ubiquitin chains. One reason for the inefficient removal of ubiquitin conjugates might be the formation of aggregates and a limited access of the linkage-specific DUBs to the corresponding isopeptide bonds. Increasing the amount of recombinant Cezanne (OTU domain) and OTUB1 (full-length) did not improve the cleavage efficiency.

We agree that this is not very convincing at the moment, and have moved the figure into the Expanded View section (Fig. E3D).

Referee #2:

This paper describes a role for the deubiquitylating enzyme Cezanne, in inhibiting the degradation of HIF-1alpha and probably HIF-2alpha. It is proposed that this occurs by removal of ubiquitin lysine 11 linkages, which otherwise promote destruction of HIF-1alpha by a proteasome independent pathway. It is also proposed that this process is dependent on VHL but independent of hydroxylation.

If true this would be an interesting finding revealing further complexity in pathways that regulate HIF-1alpha by degradation.

Points to be considered or clarified:

(i) The finding that the pathway is dependent on VHL but independent of hydroxylation is odd since known VHL functions on the HIF pathway are hydroxylation dependent. The data presented in support of this statement is not altogether convincing. The data presented in lanes 1 and 2 of figure 4C is critical. From the illustration provided it appears possible that there is some reduction in the HIF-1alpha signal. How many times was this experiment repeated? Careful repetition is required to be sure of this point. Since this is an important (and unexpected) aspect of the report the authors should perform additional experiments. Intervention in other RCC cells, effects on HIF-2alpha and effects on transfected prolyl mutated forms of HIF-1/2alpha (which should not be recognized by VHL) should all be readily possible and should be capable of confirming or refuting this interpretation.

The experiment mentioned above (Fig. 4E) was repeated four times. Quantification using ImageJ software suggested that in VHL-negative RCC4 cells HIF-1 α levels are equal in control and Cezanne knockdown cells, whereas in HA-VHL expressing RCC4 cells HIF-1 α levels are decrease (by approx. 25%) when Cezanne is depleted (Fig. 4E and E4E). In our revised manuscript we also included data showing that in the VHL-negative renal cell carcinoma cell lines A498, loss of Cezanne had no effect on HIF-1 α levels either (Fig. 4F), further supporting our hypothesis that Cezanne regulates HIF-1 α in a pVHL-dependent manner. Indeed it is surprising that at the same time hydroxylase activity seemed to be dispensable, especially because known pVHL functions on the HIF pathway are hydroxylase dependent as reviewer #2 mentioned. One explanation could be that pVHL acts as a scaffold in our scenario, helping to place Cezanne in close proximity to HIF-1 α . But this has to be analyzed in more detail.

As reviewer #2 suggested we also obtained a HIF-1 α proline mutant construct. However, we were not able to generate consistent data by overexpressing HA-tagged wt or mutant HIF-1 α together with siRNA oligonucleotides. We hope we still convinced reviewer #2 that the results in the RCC cell lines allow concluding that pVHL plays a role in HIF-1 α regulation by Cezanne.

(ii) Figure 3F is not entirely convincing. How many times was this experiment repeated?

Please see response to reviewer #1 (minor issues point 4). We have moved this figure into the Expanded View section (Fig. E3D).

(iii) *The authors should consider intervening on UBE2S and measuring the effect on HIF-1alpha stability to reinforce their conclusions.*

We followed reviewer #2's advice and co-depleted Cezanne and UBE2S (Fig. E3E). UBE2S was reported to associate with pVHL and to target the tumour suppressor for ubiquitin-mediated proteolysis (Jung *et al.*, 2006), thereby stabilizing HIF-1 α . Our data suggest that Cezanne's effect on HIF-1 α does not depend on the E2 enzyme UBE2S. HIF-1 α protein levels were decreased to the same extent in Cezanne knockdown cells and in Cezanne and UBE2S double-knockdown cell.

As was shown by the Rape lab and other labs, UBE2S is cell cycle regulated, and its roles with APC/C are now well established. It will be important to understand which E2 and E3 enzymes mediate Lys11-linked ubiquitin chain assembly on HIF1 α , but this goes beyond the scope of this manuscript.

Referee #3:

Cezanne (also known as OTU7B) is a deubiquitinating enzyme shown in previous work to disassemble K11-linked polyubiquitin preferentially. This submission from Bremm et al. reports the very interesting observation that Cezanne contributes to the regulation of HIF-1 α , an important transcription factor. Data are provided that show that loss of Cezanne destabilizes HIF-1 α , and that, surprisingly, the HIF-1 α degradation that results is proteasome independent. These all are strengths of the paper. However, there are several problems. As elaborated below, some of the paper's claims are only weakly supported, the mechanism that links Cezanne to HIF-1 α - a key aspect of the study's significance - was investigated rather superficially, and the quality of some of the experimental data is disappointing.

1. The claim that p97 is involved in HIF-1 α degradation (see Abstract and pages 4 & 12) is based on very superficial observations. Because p97 has so many diverse functions (e.g., membrane fusion & trafficking, gene expression, and both proteasomal and lysosomal degradation pathways), the effects of p97 knockdown on HIF-1 α levels as described by Bremm et al. could be very indirect. Without evidence from more detailed studies, there is no reason to think that p97 is involved directly in HIF-1 α degradation.

We agree with reviewer #3, and this section has been reworded. Fig. 4A in our revised manuscript demonstrates that co-depletion of Cezanne and p97 rescued decreased HIF-1 α protein levels. We do not claim that Cezanne's effect on HIF-1 α depends on p97 anymore and we acknowledge that the observed rescue could well be mediated by an indirect effect. Since p97 regulates many degradation pathways in the cell, we only hypothesize on the basis of Fig. 4A that Cezanne regulates HIF-1 α protein degradation. This assumption is then further supported by our results shown in Fig. 4 (especially 4G & 4H).

2. Despite providing evidence that Cezanne disassembles K11-polyubiquitin, that Cezanne and HIF-1 α can co-IP, and that immunoprecipitated HIF-1 α contains K11-linked polyubiquitin, the authors never showed that Cezanne specifically removes K11-linked polyubiquitin from HIF-1 α conjugates (see #3, below). They certainly imply that Cezanne has that function, although their model (Fig 4G) is rather vague on this point. This is a key issue raised by the authors' observations and, as the tools appear to be available to investigate it, should be examined experimentally.

To address this specific question we used ubiquitin chain restriction analysis on immunoprecipitated HIF-1 α (Fig. E3D). More importantly, we now also observed that depletion of Cezanne increased Lys11 linkages in precipitates obtained with an anti-HIF-1 α antibody from HeLa cells (new Fig. 3F). This suggests that Cezanne can regulate Lys11 linkages in immunoprecipitated HIF-1 α .

3. (Fig. 3F) The extent of deubiquitination by Cezanne or OTUB1 of HIF-1 α conjugates is difficult to gauge from the western blot shown. The Cezanne digestion in particular appears to have had a modest effect. Presumably, detection used an anti-ubiquitin antibody. The experiment would be much more informative if the blots were also developed with anti-K11 and anti-K48 antibodies (or,

that Lys11-linked ubiquitin chains are regulated by changes in oxygen levels in the cell, we have removed the above-mentioned sentences.

8. (p. 12, 1st para) *The sentence "Our data suggest that depletion of p97..." isn't clear*

Please see our response to your first concern raised regarding our interpretation of the Cezanne and p97 co-depletion experiment. We have changed the entire section now and removed the sentences mentioned above.

2nd Editorial Decision

15 September 2014

Thank you for the submission of your revised manuscript to EMBO reports. We have now received the enclosed reports from the three referees that also saw the previous version. As you will see, all referees are now much more supportive of publication, although a few minor issues need to be addressed.

Referee points out a minor mistake in the text, and referee 2 requests that the the limitations of your study in understanding why there is a requirement for pVHL but not hydroxylation be explicitly discussed, and the experiments that proved unsuccessful be mentioned. Referee 3 mentions a minor issue regarding the precision used in Expanded figure 4H, but also raises an important concern regarding the in vitro linkage-selective digestions. In this context, I wonder if it would be feasible to do a double digestion with OTUB1 and Cezanne in a relatively short time frame, to complement this experiment, which could then be included in the main text. Please let me know if you think this is a feasible and reasonable option.

Referee reports

Referee #1:

The authors have responded well to my earlier critique, and the paper is ready for publication. One minor issue: the CMA mutant should be Fig. 4I, not E4I.

Referee #2:

The authors have provided additional data that addresses my question on the security of the proposed 'VHL-independence' of this response and on the effect UBE2S depletion. They state that they were not able to obtain consistent data by overexpressing wild-type or proline mutant HIF-1alpha.

The dependence on VHL is puzzling.

I am quite happy that findings are published without clear understanding (which is the case), but it is important to highlight what is not known and significant experiments that have been attempted without success.

I therefore think that the authors should do precisely that.

i.e. state clearly that

'in view of known dependence of pVHL binding to HIF on prolyl hydroxylation, the apparent dependence of the action of Cezanne on pVHL, but not hydroxylation, is unexplained. Attempts to test the action of Cezanne on prolyl mutant HIF-1alpha proteins have so far been unsuccessful'

On this basis I think the manuscript would be suitable for publication.

Referee #3:

The revised manuscript from Bremm et al., while greatly improved, failed to bolster weak evidence in the original paper that HIF-1 α is modified with both K48 and K11-linked polyubiquitin. Instead, the inconclusive results of the original Fig 3F are now relegated to the Expanded View (i.e., supplementary) figures. This is disappointing, as I consider the issue addressed by the original Fig 3F to be central to the study. I am perplexed as to why the authors have not provided a clearer analysis using their original approach (i.e., digestions by linkage-selective DUBs of immunoprecipitated HIF-1 α), and why OTUB1 and Cezanne were not combined for one of the digestion samples, as recommended. Whereas some data (see Figs 3E & 3F) are consistent with the proposal that HIF-1 α is modified by both K11 and K48 chains, the possibility that other ubiquitinated proteins are in the anti-HIF-1 α IP's makes those results inconclusive.

A second, minor point is that the values of % cells in different phases of the cell cycle in Fig. E4H are presented with unrealistic precision (i.e., to two decimal places).

2nd Revision - authors' response

24 September 2014

Many thanks for your e-mail and your comments, I have uploaded the revised files of our manuscript "Cezanne (OTUD7B) regulates HIF-1 α homeostasis in a proteasome-independent manner" through the EMBO reports website now. We have addressed the points raised by the referees and yourself as specified below:

- 1) We amended the reference to figure 4I in the main text.
- 2) We included a short discussion on the HIF-1 α Pro-mutants as suggested by referee #2. (*"This suggests that while Cezanne does not affect pVHL levels (Fig 2F), it regulates HIF-1 α homeostasis in a pVHL-dependent way. HIF-1 α hydroxylation is a prerequisite for pVHL-mediated HIF-1 α degradation via the proteasome. The apparent dependence of Cezanne on pVHL, but not hydroxylation, is unexplained. Attempts to test the action of Cezanne on prolyl mutant HIF-1 α have so far been unsuccessful."*)
- 3) Referee #3 pointed out that figure E4H is presented with unrealistic precision. Therefore, we rounded all values to integral numbers.
- 4) The main issue raised by referee #3 is figure E3D. Our explanation for the inefficient cleavage of ubiquitin chains on HIF-1 α by Cezanne and OTUB1 is that immunoprecipitated HIF-1 α from HeLa cells may be partly aggregated. While the highly promiscuous USP21 was still able to remove all ubiquitin chains from HIF-1 α , precipitated HIF-1 α generated problems for Cezanne and OTUB1. It is known that heavily ubiquitinated proteins are prone to form aggregates that are difficult to resolve, and we think that the linkage specific DUBs Cezanne and OTUB1 are less efficient in processing these aggregates (because they cannot 'get in'). We do not know how complex ubiquitin chains are that are attached to HIF-1 α , and additional well-known post-translational modifications on HIF-1 α may also interfere with efficient cleavage of polyubiquitin conjugates by Cezanne and OTUB1. We have done the ubiquitin chain restriction analysis (UbiCREST) on HIF-1 α several times, but unfortunately cannot provide a better figure at the moment. While UbiCREST is still under development in the Komander laboratory in Cambridge, work on the role of atypical ubiquitination in HIF-1 α regulation is now continued in Frankfurt and Dundee. This makes logistics of further repeats of the experiment quite challenging and not feasible at the moment. However, using the linkage specific antibodies on precipitated HIF-1 α from cells clearly showed that both K48- and K11-linked ubiquitin chains are present on HIF-1 α . The additional UbiCREST experiment would not add anything to the paper at the moment.

Thank you very much again for your support during the revision process. I look forward to hear

from you in due course.

3rd Editorial Decision

29 September 2014

I am very pleased to accept your manuscript for publication in the next available issue of EMBO reports.

Thank you again for your contribution to EMBO reports and congratulations on a successful publication.